# Health system interventions for adults with type 2 diabetes in low- and middle-income countries: A systematic review and meta-analysis

David Flood[1,2]*, Jessica Hane[3], Matthew Dunn[4], Sarah Jane Brown[5], Bradley H. Wagenaar[6,7], Elizabeth A. Rogers[8,9], Michele Heisler[10,11,12], Peter Rohloff[1], Vineet Chopra[12,13]

1 Center for Research in Indigenous Health, Wuqu' Kawoq, Tecpán, Guatemala, 2 Division of Hospital Medicine, Department of Internal Medicine, National Clinician Scholars Program, University of Michigan, Ann Arbor, Michigan, United States of America, 3 Medicine-Pediatrics Residency Program, University of Minnesota, Minneapolis, Minnesota, United States of America, 4 School of Public Health, University of Michigan, Ann Arbor, Michigan, United States of America, 5 Health Sciences Libraries, University of Minnesota, Minneapolis, Minnesota, United States of America, 6 Department of Global Health, University of Washington, Seattle, Washington, United States of America, 7 Department of Epidemiology, University of Washington, Seattle, Washington, United States of America, 8 Division of General Internal Medicine, Department of Medicine, University of Minnesota, Minneapolis, Minnesota, United States of America, 9 Department of Pediatrics, University of Minnesota, Minneapolis, Minnesota, United States of America, 10 Department of Internal Medicine, University of Michigan, Ann Arbor, Michigan United States of America, 11 Institute for Healthcare Policy and Innovation, University of Michigan, Ann Arbor, Michigan United States of America, 12 Center for Clinical Management Research, Veterans Affairs Ann Arbor Healthcare System, Ann Arbor, Michigan United States of America, 13 Division of Hospital Medicine, Department of Medicine, University of Michigan, Ann Arbor, Michigan United States of America

* david@wuqukawoq.org

**Data Availability Statement:** The study's dataset and statistical code are available through Dataverse at: https://doi.org/10.7910/DVN/NIESKT.

## Abstract

### Background

Effective health system interventions may help address the disproportionate burden of diabetes in low- and middle-income countries (LMICs). We assessed the impact of health system interventions to improve outcomes for adults with type 2 diabetes in LMICs.

### Methods and findings

We searched Ovid MEDLINE, Cochrane Library, EMBASE, African Index Medicus, LILACS, and Global Index Medicus from inception of each database through February 24, 2020. We included randomized controlled trials (RCTs) of health system interventions targeting adults with type 2 diabetes in LMICs. Eligible studies reported at least 1 of the following outcomes: glycemic change, mortality, quality of life, or cost-effectiveness. We conducted a meta-analysis for the glycemic outcome of hemoglobin A1c (HbA1c). GRADE and Cochrane Effective Practice and Organisation of Care methods were used to assess risk of bias for the glycemic outcome and to prepare a summary of findings table. Of the 12,921 references identified in searches, we included 39 studies in the narrative review of which 19 were cluster RCTs and 20 were individual RCTs. The greatest number of studies were conducted in the East Asia

**Funding:** DF is supported by the National Clinician Scholars Program at the University of Michigan Institute for Healthcare Policy & Innovation. BHW is supported by grant number K01MH110599 from the National Institute of Mental Health. EAR is supported by the National Institute of Diabetes and Digestive and Kidney Diseases of the National Institutes of Health under award number K23DK118207. The funders had no role in study design, data collection and analysis, decision to publish, or preparation of the manuscript.

**Competing interests:** The authors have declared that no competing interests exist.

**Abbreviations:** EPOC, Effective Practice and Organisation of Care; HbA1c, hemoglobin A1c; HICs, high-income countries; LMICs, low- and middle-income countries; RCT, randomized controlled trial; SMS, short message service.

and Pacific region ($n = 20$) followed by South Asia ($n = 7$). There were 21,080 total participants enrolled across included studies and 10,060 total participants in the meta-analysis of HbA1c when accounting for the design effect of cluster RCTs. Non-glycemic outcomes of mortality, health-related quality of life, and cost-effectiveness had sparse data availability that precluded quantitative pooling. In the meta-analysis of HbA1c from 35 of the included studies, the mean difference was −0.46% (95% CI −0.60% to −0.31%, $I^2$ 87.8%, $p < 0.001$) overall, −0.37% (95% CI −0.64% to −0.10%, $I^2$ 60.0%, $n = 7$, $p = 0.020$) in multicomponent clinic-based interventions, −0.87% (−1.20% to −0.53%, $I^2$ 91.0%, $n = 13$, $p < 0.001$) in pharmacist task-sharing studies, and −0.27% (−0.50% to −0.04%, $I^2$ 64.1%, $n = 7$, $p = 0.010$) in trials of diabetes education or support alone. Other types of interventions had few included studies. Eight studies were at low risk of bias for the summary assessment of glycemic control, 15 studies were at unclear risk, and 16 studies were at high risk. The certainty of evidence for glycemic control by subgroup was moderate for multicomponent clinic-based interventions but was low or very low for other intervention types. Limitations include the lack of consensus definitions for health system interventions, differences in the quality of underlying studies, and sparse data availability for non-glycemic outcomes.

## Conclusions

In this meta-analysis, we found that health system interventions for type 2 diabetes may be effective in improving glycemic control in LMICs, but few studies are available from rural areas or low- or lower-middle-income countries. Multicomponent clinic-based interventions had the strongest evidence for glycemic benefit among intervention types. Further research is needed to assess non-glycemic outcomes and to study implementation in rural and low-income settings.

## Author summary

### Why was this study done?

- Approximately 80% of the 463 million adults with type 2 diabetes worldwide live in low- and middle-income countries (LMICs).

- Evidence-based treatments for diabetes exist, but health systems in LMICs have difficulty meeting diabetes patients' needs.

- Health system interventions can help address this gap by improving the delivery of diabetes care within health systems.

### What did the researchers do and find?

- We conducted a systematic review and meta-analysis of 39 health system interventions aiming to improve outcomes of glycemic (i.e., blood glucose) control, mortality, quality of life, or cost-effectiveness for people with type 2 diabetes in LMICs.

- We found that health system interventions for type 2 diabetes may be effective in improving glycemic control in LMICs, but few studies were available from rural areas or low- or lower-middle-income countries.

- Among intervention types, multicomponent clinic-based interventions had the strongest evidence for improving glycemic control.

### What do these findings mean?

- Our findings support the scaling up of diabetes health system interventions to improve patients' glycemic control in LMICs.

- Further research is needed to assess other outcomes beyond glycemic control, especially in rural areas and in low- or lower-middle-income countries.

## Introduction

Type 2 diabetes disproportionately affects people in low- and middle-income countries (LMICs). Of the estimated 463 million adults worldwide with type 2 diabetes, approximately 80% reside in LMICs [1]. The absolute number of adults and percentage of the population with diabetes have increased more quickly in LMICs than in high-income countries (HICs) [2]. Despite the existence of cost-effective and evidence-based clinical treatments for type 2 diabetes [3], health systems in LMICs have difficulty meeting the rising need for quality care [4]. Improving and scaling up care in LMICs is an urgent global health priority.

Health system interventions can help address this priority. In contrast to clinical therapies for individual patients, health system interventions emphasize the behavior of health workers and the way healthcare is practiced and delivered [5]. Examples of health system interventions include quality and safety programs, health information systems, health worker incentives, and changes in scope of practice [5]. Effective health system interventions are needed to implement type 2 diabetes care in settings with different resources, cultures, and population risk factors [6].

While health system interventions improve type 2 diabetes outcomes in HICs [7–9], the evidence from LMICs is limited. A 2012 meta-analysis of 142 randomized trials primarily conducted in HICs found that interventions targeting the health system rather than healthcare providers or patients alone were most effective [9]. However, health system interventions designed and tested in HICs may not be generalizable to LMICs [10]. In LMICs, prior reviews draw from diverse study designs and together suggest a modest yet increasing number of studies on the implementation of evidence-based type 2 diabetes care into health systems in LMICs [11–13]. To our knowledge, no review has systematically assessed evidence from randomized controlled trials (RCTs) or conducted a meta-analysis.

Therefore, we conducted a systematic review and meta-analysis to examine the impact of health system interventions that aimed to improve outcomes of glycemic (i.e., blood glucose) change, mortality, health-related quality of life, or cost-effectiveness for adults with type 2 diabetes in LMICs.

## Methods

This systematic review and meta-analysis was conducted based on guidance from Cochrane Effective Practice and Organisation of Care (EPOC), a group focusing on reviews of the

delivery of health services [14]. We registered the review in PROSPERO (CRD42018106765; S1 Appendix) and followed the Preferred Reporting Items for Systematic Reviews and Meta-analyses (PRISMA) guidelines (S2 Appendix) [15]. Ethical approval was not required as the research used publicly available data.

## Search strategy and selection criteria

We performed systematic searches in several bibliographic databases. The search strategy was built and tested for sensitivity in Ovid MEDLINE (S3 Appendix) and translated to 5 other bibliographic databases: Cochrane Library, EMBASE, African Index Medicus, LILACS, and Global Index Medicus. Databases were chosen to be inclusive of international and interdisciplinary literature. The search strategy was built in English, and no language filters were applied. We also hand-searched the references of included studies, related systematic reviews, and the websites of major international diabetes organizations. To ensure high search quality, a second reference librarian peer-reviewed the search terms. The search dates were from database inception through February 24, 2020.

We included RCTs of health system interventions targeting non-pregnant, ambulatory adults with type 2 diabetes in LMICs. We defined LMICs using the 2019 World Bank income groups. Included studies reported at least 1 of the following outcomes: glycemic change, mortality, health-related quality of life, or cost-effectiveness. Given our interest in durable health system interventions, we prespecified that studies enroll 100 or more participants, with follow-up of at least 24 weeks. No date or language restrictions were applied.

We used the EPOC review group's definition of health system interventions as those designed to "improve the professional practice and the delivery of effective health services" through changes in healthcare delivery, financing, governance, and implementation [5,14]. Consistent with EPOC, we excluded studies of patient behavior change alone if the intervention did not primarily target healthcare professionals [14]. For example, an intervention training healthcare professionals on diabetes education was included; however, an intervention aiming to improve outcomes solely through individualized diabetes education was excluded [14]. We defined "healthcare professional" broadly to encompass physicians, nurses, pharmacists, and other allied health workers.

## Data analysis

A medical librarian (SJB) downloaded all records, removed duplicates, and imported records to the review management tool Covidence. Two authors (DF and JH) independently screened studies by title and abstract and, subsequently, by full-text review. Disagreements were resolved first by consensus and, if needed, in consultation with another author (PR). We used language proficiency among the members of our review team and Google Translate to abstract data from non-English trials [16]. Multiple reports from the same study were identified by reviewing the country setting, intervention details, and authorship list. When multiple reports were identified, we linked the reports together for extraction and analysis. We used the TIDieR checklist and EPOC template to structure extraction [17]. We extracted study elements including the 4 outcomes, country, setting, duration and follow-up, number of participants enrolled, intervention description, and comparator. One author (DF) extracted summary data into a customized electronic spreadsheet, and 2 other authors (JH and MD) independently verified the extracted data. We classified each study by EPOC domain (S4 Appendix) [5] and then grouped interventions into similar types. Our main unit of analysis was at the level of intervention type. If outcomes were missing or not reported, we contacted authors twice to obtain data. We used GRADE and EPOC guidance to assess risk of bias for the glycemic outcome and to prepare a summary of findings table [18–21].

## Statistical analysis

As quantitative data were reliably reported for only 1 of our 4 included outcomes, we limited our meta-analysis to the glycemic outcome of hemoglobin A1c (HbA1c) change. The meta-analysis was performed with random effects using the DerSimonian–Laird method for mean between-group HbA1c difference. Prespecified subgroup analyses were done by intervention type. Sample sizes for cluster RCTs were adjusted to account for the design effect using the intracluster correlation coefficient (ICC) [22]. We inferred an ICC from the literature if one was not reported in the study or its trial protocol [23]. We followed the methodology recommended in the Cochrane handbook to calculate within-group mean and standard deviation when this information was not directly reported in the study or made available by authors [22]. To provide a range of the effects of individual studies, we calculated an overall prediction interval [24].

We conducted 2 sensitivity analyses. First, we excluded studies with high risk of bias. Second, we assessed the influence of individual studies by using the leave-one-out method to recalculate estimates omitting 1 study at a time [25]. Heterogeneity was explored by calculating $I^2$ and $T^2$, and we report 95% confidence intervals for $I^2$ if 3 or more studies are pooled [26]. Publication bias was assessed by visual inspection of funnel plots and the Egger test. The trim-and-fill method was also applied to impute the number of studies potentially missing from the meta-analysis and to re-estimate an overall effect size accounting for publication bias [27]. We analyzed data in Stata (version 16.0).

## Results

### Overview of results

Our search strategy identified 12,921 references (Fig 1). After removing 1,093 duplicates, we screened 11,828 references by title and abstract and assessed 322 full-text articles for eligibility. Of the 283 articles excluded after full-text review, 103 articles were excluded due to the type of intervention, and 94 articles were excluded due to incomplete data. We included 39 trials in the narrative review and 35 trials in the meta-analysis of glycemic change.

Of the 39 studies included in the narrative review, 19 were cluster RCTs and 20 were individual RCTs (Table 1; S5 Appendix). There were 21,080 total participants enrolled across included studies. The greatest number of studies were conducted in the East Asia and Pacific region, followed by South Asia. Twenty-nine studies were conducted in upper-middle-income countries as defined by the World Bank, and only 1 trial included a site in a low-income country. All but 1 of the studies were published in the year 2010 or after [28]. The study setting was primarily urban in 27 trials and primarily rural in 5 trials. The median study duration was 10 months (interquartile range 6 to 12). Most interventions involved the EPOC domains of delivery arrangements and implementation strategies. Only 1 intervention incorporated a change in governance [29], and no study tested changes in financial arrangements. The comparator group in most studies was usual care as defined by the local program or setting of care. Two studies described the comparator group as enhanced usual care, where the enhancement consisted of clinical training for health professionals [30,31], and in 1 study the medical fees were waived in the comparator arm [32].

### Narrative description of interventions

**Multicomponent clinic-based interventions.** Eight trials were classified as clinic-based multicomponent interventions, which we defined as studies involving multiple types of health workers implementing a bundle of quality improvement or health system strengthening

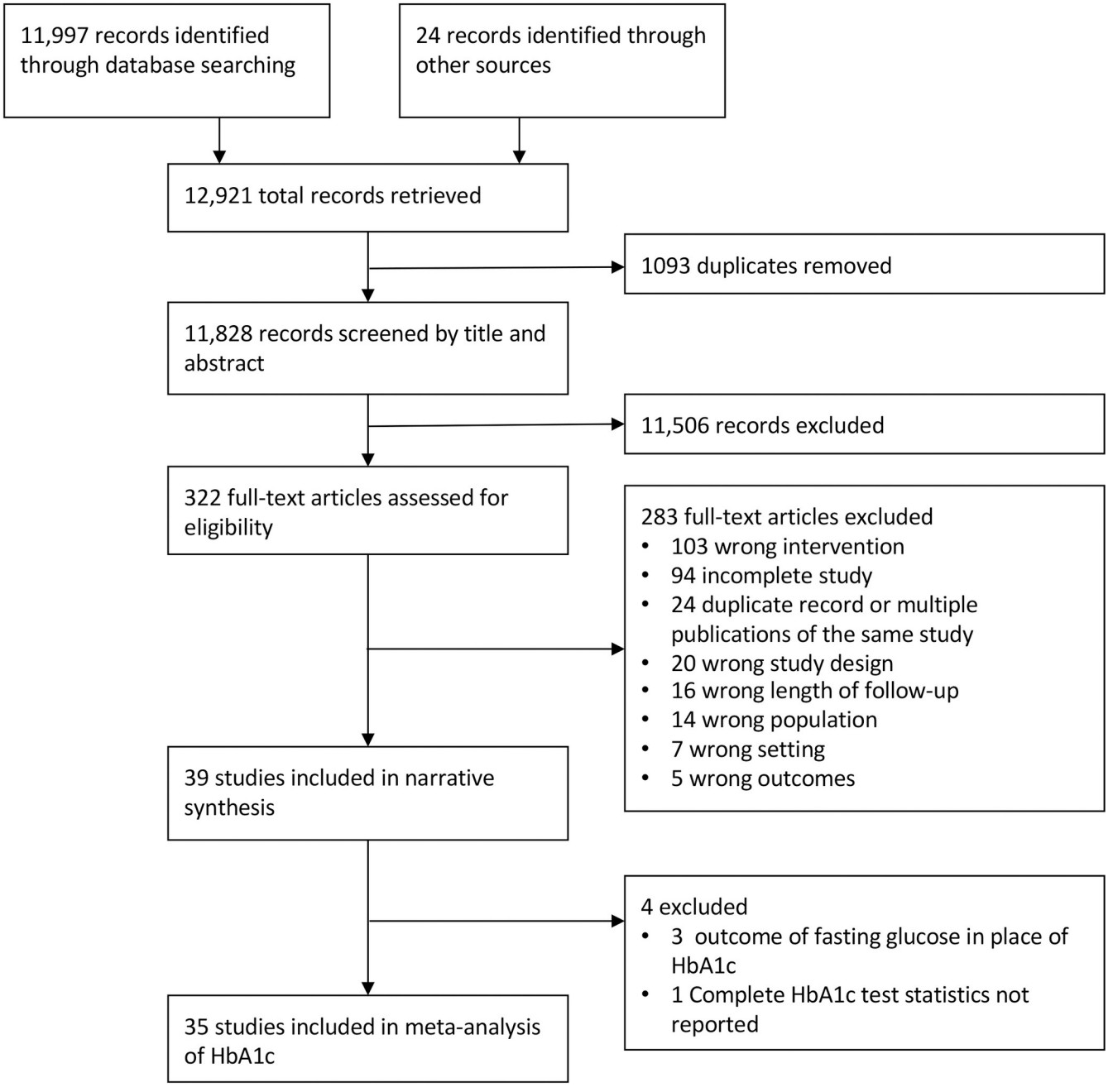

**Fig 1. PRISMA study flow diagram.** HbA1c, hemoglobin A1c.

interventions [30,31,58–63]. Most studies incorporated primary care doctors in a team-based intervention [30,31,59,60,62,63], and the study by Ali and colleagues incorporated endocrinologists [58]. Each intervention included self-management education or support delivered by peers [59], non-physician care coordinators [58], clinicians [30,59,60,62,63], or an automated short message service (SMS) text-messaging system [31]. Other components in the bundles included health record establishment [61], electronic decision support [31,58], physician education [30,59,60,62,63], care coordination or case management [58,62], clinical information

**Table 1. Characteristics of the 39 studies included in this review.**

| Characteristic | Number of studies (*n* = 39) | References |
|---|---|---|
| **EPOC domains** | | |
| Delivery arrangements | 27 | [28,32–57] |
| Delivery arrangements and implementation strategies | 9 | [30,31,58–64] |
| Implementation strategies | 2 | [65,66] |
| Delivery arrangements, governance arrangements, implementation strategies | 1 | [29] |
| **Intervention type** | | |
| Multicomponent clinic-based | 8 | [30,31,58–63] |
| Pharmacist task sharing | 14 | [28,33–35,37,38,41,42,47,48,50,51,55,57] |
| Diabetes education or support alone | 9 | [32,39,40,43,46,49,53,54,56] |
| Case management by nurses | 2 | [36,52] |
| Physician clinical training alone | 2 | [65,66] |
| Nurse task sharing | 1 | [29] |
| mHealth screening and quality improvement | 1 | [64] |
| Internet-based glucose telemonitoring alone | 2 | [44,45] |
| **Study design** | | |
| Individual RCT | 20 | [28,33–35,37–39,41,42,44,47,48,50–53,55,58,59,61] |
| Cluster RCT | 19 | [29–32,36,40,43,45,46,49,54,56,57,60,62–66] |
| **World Bank region**[*] | | |
| East Asia and Pacific | 20 | [28,32,34–37,39,40,44,45,49,52–54,57,61–63,66] |
| South Asia | 7 | [30,31,38,42,43,51,58] |
| Latin America and Caribbean | 4 | [47,48,59,60] |
| Sub-Saharan Africa | 4 | [29,33,46,53] |
| Middle East and North Africa | 4 | [41,50,55,64] |
| Europe and Central Asia | 1 | [65] |
| **World Bank income group**[*] | | |
| Low | 1 | [53] |
| Lower middle | 11 | [30,31,33,38,42,43,49,51,53,58,66] |
| Upper middle | 29 | [28,29,32,34–37,39–41,44–48,50,52,54–57,59–66] |
| **Setting** | | |
| Mostly rural | 5 | [30,31,36,49,64] |
| Mostly urban | 27 | [29,32,33,35,37–42,44,46,48,50,52,54–62,65,66] |
| Mixed | 4 | [43,45,53,63] |
| Not reported | 3 | [28,34,47] |
| **Outcome reported** | | |
| Mortality | 19 | [29–31,34,36,39,42,44,45,49,51–53,57–59,61,63,66] |
| Health-related quality of life | 11 | [29,32,33,40,45,46,51,52,58,59,62] |
| Cost-effectiveness | 5 | [33,36,46,58,59] |
| Change in glycemic control | 39 | All included studies |

[*]The studies by Van Olmen et al. [53] and Reutens et al. [66] are counted twice, as they were conducted in multiple countries of different World Bank regions and income groups.

EPOC, Effective Practice and Organisation of Care; RCT, randomized controlled trial.

systems [30,60,62,63], and clinical audit and feedback [62,63]. Three studies were based on the Chronic Care Model [60,62,63].

**Pharmacist task-sharing interventions.** Fourteen studies were classified as pharmacist task-sharing interventions, which we defined as studies in which patients received activities

performed by pharmacists such as care coordination, medication review and counseling, and prescription suggestions to physicians [28,33–35,37,38,41,42,47,48,50,51,55,57]. All pharmacist task-sharing interventions incorporated diabetes self-management education, but no trial included education alone. Seven interventions included counseling and reminders through telephone calls or text messages [34,35,38,39,41,55,57], and a study in Iran incorporated only telephone calls with no face-to-face encounters [50]. No intervention involved pharmacists independently prescribing or titrating medications. All 11 of the studies that described the study setting were conducted in an urban area [33,35,37,38,41,42,48,50,51,55,57]. The intensity of interventions was incompletely reported but ranged from 3 to 16 telephone calls or face-to-face visits.

**Diabetes education or support alone.** Nine studies involved health workers primarily implementing diabetes education or support without additional services [32,39,40,43,46,49,53,54,56]. We defined these interventions as diabetes education or support alone. Six of the studies primarily involved in-person delivery [32,40,43,46,49,56], and 3 studies delivered the intervention in group format [46,49,56]. The health workers in these studies varied between and within studies and included peers [49,53,56], community health workers [43,46,53], nurses [32,39,40,53,54], psychologists [32], and physicians [32,39,54]. Face-to-face encounters were supplemented with telephone calls in 2 studies [32,54] and by computer-assisted instruction in another study [40]. Motivational interviewing techniques were incorporated in 2 studies [32,46]. The intensity of the in-person interventions ranged from 4 to 24 total face-to-face encounters.

**Other intervention types with fewer studies.** Two studies involved nursing case management interventions. In these trials, a nurse [52] or nurse–community health worker team [36] facilitated patient support and care coordination. Both trials varied intervention intensity by a patient's risk factors. DePue and colleagues conducted a cluster RCT in American Samoa that primarily used home visits and individual rather than group sessions [36]. In the trial conducted by Tutino and colleagues in China, both the intervention and comparator arms included implementation of a web-based clinical information portal, and the intervention arm received additional nurse-led care coordination [52].

Two health system interventions involved physician clinical training alone [65,66]. Akturan and colleagues trained physicians on a therapeutic interviewing technique [65]. Reutens and colleagues trained physicians in multiple countries on diabetes guidelines using 2 in-person sessions and reminders [66].

One study was classified as a nurse task-sharing intervention [29]. Conducted in South Africa, this intervention involved authorizing, training, and supporting nurses to independently prescribe a set of drugs for several noncommunicable diseases including diabetes using an algorithmic management tool [29].

One study was classified as an mHealth screening and quality improvement intervention [64]. This trial involved an intervention for diabetes and hypertension involving SMS educational messages and appointment reminders, community-based screening, and deployment of electronic clinical tools for physicians and nurses [64].

Two studies tested glucose telemonitoring interventions in which participants uploaded glucose data to an online system and then received feedback from health workers regarding treatment changes to improve glycemic control [44,45].

## Summary of outcomes

We describe outcomes of glycemic change, mortality, quality of life, and cost-effectiveness by study in S5 Appendix. Glycemic changes were reported based on HbA1c values in 36 studies

and based on fasting glucose alone in 3 trials. Among studies reporting fasting glucose only, 2 trials of multicomponent clinic-based interventions found glycemic improvement [56,61], while there was no improvement in a trial of diabetes education or support alone [43]. The primary outcome involved change in HbA1c or the proportion of participants meeting HbA1c goals in 23 studies.

Outcomes of mortality, health-related quality of life, and cost-effectiveness were reported in 19, 11, and 5 studies, respectively. Of the 19 studies reporting mortality, 14 studies had 10 or fewer deaths combined in the intervention and comparator groups (S6 Appendix). Studies with larger numbers of deaths appeared to have generally similar mortality between trial arms though a formal meta-analysis was not conducted due to sparseness of data [29,31,53,58]. No study's primary outcome was mortality.

Of the 11 studies reporting quality of life, 6 studies reported no significant differences between the intervention and comparator arms [29,32,45,46,52,59], and 5 studies showed improved quality of life in the intervention arm [33,40,51,58,62]. Seven different scales were used to assess quality of life, and only the EuroQol EQ-5D was used in more than 1 study [29,45,46,52,58]. Only 1 study reported quality of life as a primary outcome [51].

Cost-effectiveness was reported as an incremental cost-effectiveness ratio (ICER) in 5 studies. An ICER of \$1,121 per 1% decrease in HbA1c was reported in the trial by DePue and colleagues [36] and \$1,850 in the study by Ali and colleagues [58]. The study by Mash et al. reported an ICER of \$1,862 per quality-adjusted life year (QALY) based on improvements in blood pressure [46]. Two other trials calculated an ICER between trial arms [33,59]. No study reported cost-effectiveness as a primary outcome.

In the meta-analysis of HbA1c in 35 trials, there were 10,060 total participants when accounting for the design effect of cluster RCTs (5,240 in intervention arms and 4,820 in comparator arms). The overall between-arm HbA1c mean change was −0.46% (95% CI −0.60% to −0.31%, $I^2$ 87.8% [95% CI 84.0% to 90.6%]; Fig 2). Within subgroups of intervention type, mean HbA1c difference was −0.37% (95% CI −0.64% to −0.10%, $I^2$ 60.0% [95% CI 8.2% to 82.6%], $n = 7$) in multicomponent clinic-based interventions, −0.87% (95% CI −1.20% to −0.53%, $I^2$ 91.0% [95% CI 86.5% to 94.0%], $n = 13$) in pharmacist task-sharing studies, and −0.27% (95% CI −0.50% to −0.04%, $I^2$ 64.1% [95% CI 18.8% to 84.1%], $n = 7$) in trials of diabetes education or support alone. The effect sizes of other intervention types with 2 or fewer studies reporting HbA1c are summarized in Fig 2. The overall HbA1c prediction interval was −1.19% to 0.28%.

Studies are listed in the figure by first author [28–36,38–42, 44–55,57–60,62–66]. The intervention arms were combined in the study by Anzaldo-Campos and colleagues [59]. Only the health literacy intervention arm was included in the study by Wang and colleagues [54]. The proportion of participants from low- and middle-income countries was inferred to be 60% in the study by Reutens and colleagues [66]. Participant numbers in cluster RCTs are adjusted for design effect as described in the Methods. The prediction interval is depicted as the horizontal whiskers intersecting the overall effect diamond marker.

## Risk of bias and sensitivity analysis

Eight studies were at low risk of bias for the summary assessment of glycemic control, 15 studies were at unclear risk, and 16 studies were at high risk (S7 Appendix). The overall funnel plot and Egger test for the HbA1c meta-analysis suggested possible bias (Egger $p < 0.001$; S8 Appendix), but there was little evidence of bias within subgroups of intervention types (S9 Appendix). Using the trim-and-fill method, we estimated that there were 8 missing studies, and inclusion of these imputed studies resulted in an estimated overall HbA1c mean difference

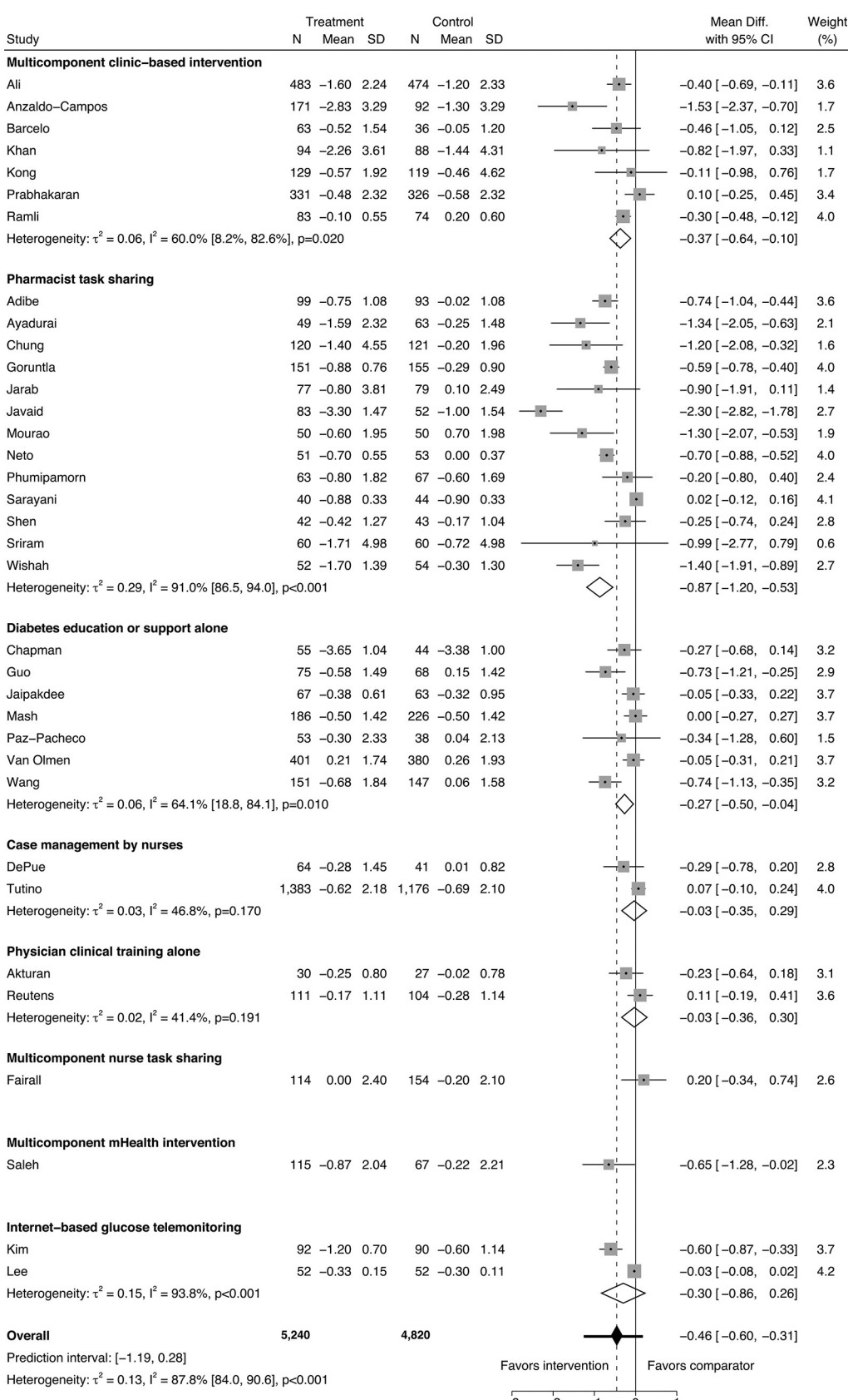

**Fig 2. Forest plot for meta-analysis of hemoglobin A1c (%) mean difference.**

of −0.28% (95% CI −0.43% to −0.13%; S10 Appendix). In the sensitivity analysis of studies not at high risk of bias ($n$ = 21 trials), the overall HbA1c mean difference was −0.20% (95% CI −0.32% to −0.08%, $I^2$ 71.8% [95% CI 56.2% to 81.8%]; S11–S13 Appendices). In the sensitivity analysis using the leave-one-out method, we found that exclusion of the study with the largest effect size [42] would result in a HbA1c mean difference of −0.39% (95% CI −0.52% to −0.26%, $I^2$ 84.5%; S14 Appendix).

The certainty of evidence using the GRADE/EPOC approach for glycemic control by subgroup was moderate for multicomponent clinic-based interventions but was low or very low for other intervention types (S15 Appendix). The most common reasons for downgrading the certainty of evidence for intervention types were concerns regarding risk of bias or inconsistency across studies (S16 Appendix). For example, in the case of pharmacist task-sharing interventions, 9 of the 14 studies were classified as being at high risk of bias, 5 were at unclear risk of bias, and none were at low risk of bias. The absence of high-quality trials resulted in a low certainty of evidence for pharmacist task-sharing studies despite their sizeable pooled HbA1c estimate in the meta-analysis. Conversely, in the case of multicomponent clinic-based interventions, only 1 of the 8 studies was deemed to be at high risk of bias, and 3 of the studies were at low risk of bias. The result was a moderate certainty of evidence for the glycemic outcome for these interventions despite a lower pooled HbA1c estimate than for the pharmacist-led studies.

## Discussion

We systematically reviewed the literature and identified 39 RCTs of health system interventions for adults with type 2 diabetes in LMICs that assessed glycemic control, mortality, health-related quality of life, or cost-effectiveness. Most included studies were conducted in upper-middle-income countries, and few studies were carried out in rural areas or low- or lower-middle-income countries. Mirroring global patterns [67], this research disparity is discordant with epidemiologic evidence showing a substantial diabetes burden in low-income countries and in rural areas of LMICs [68,69]. The EPOC domains of delivery arrangements and implementation strategies were most commonly involved in the included interventions. In the overall meta-analysis of HbA1c from 35 trials, we found that health system interventions modestly improved glycemic control on average. At the same time, the wide prediction interval overlapping 0 in the meta-analysis of HbA1c showed that there was a wide range of effectiveness across studies, and some health system interventions may not be effective in all settings. Non-glycemic outcomes of mortality, health-related quality of life, and cost-effectiveness were less frequently reported. There was considerable heterogeneity in the overall pooled analysis that was partially explained by intervention type and baseline HbA1c. Within intervention types, multicomponent clinic-based interventions had moderate evidence of glycemic benefit, but the certainty of evidence was low or very low for other intervention types.

Our review complements prior meta-analyses of studies primarily from HICs showing the benefit of systems-level quality improvement interventions on surrogate outcomes such as HbA1c, blood pressure, and cholesterol [7–9]. However, these prior reviews have included few trials outside of HICs, which limits generalizability to health systems in LMICs. In LMICs, published reviews of health system interventions for diabetes care have explored diabetes care models [13], integrated hypertension and diabetes care [12], and interventions with a lifestyle component [11]. Incorporating heterogeneous study designs, these previous reviews have surveyed the limited evidence and described various approaches that have been implemented in LMICs. Our review adds to the literature by focusing on clinical and patient-oriented outcomes from the increasing number of randomized trials conducted in these settings.

The most common intervention types we identified were multicomponent clinic-based interventions, pharmacist task-sharing interventions, and interventions of diabetes education or support alone. Multicomponent clinic-based interventions were modestly effective in improving glycemic control, with moderate certainty of evidence. At the same time, multiple well-conducted trials had null results [30,31,62]. These findings may reflect differences in participants, setting, or the implementation of different components in the bundle. Of note, the comparator arm in 2 of these well-conducted multicomponent clinic-based trials consisted of enhanced usual care [30,31], potentially causing an attenuation of effect size. In HICs, components with the largest effect sizes have been team change, patient education or patient self-management, electronic registries, and promotion of patient–provider communication [7].

Interventions focusing solely on implementing diabetes education or support within the health system also were effective in improving glycemic control, but the certainty of evidence was low. All 3 trials judged as low risk of bias had null results [32,43,46]. One potential explanation for inconsistent findings is the relatively low contact intensity of many studies. In HICs, a dose-dependent relationship has been observed between contact intensity and glycemic effectiveness, with interventions with 10 or fewer hours found to be ineffective [70]. Another consideration is that research trial infrastructure in resource-limited settings may catalyze the delivery of standard clinical care across trial arms. This revitalization of underlying care may make it difficult to detect modest differences attributable to education or support alone. A dramatic example of this effect was the Happy Life Club trial in China, in which both trial arms experienced 3.7% within-group HbA1c improvement over 18 months [32]. Both the intervention and comparator arm in this trial had out-of-pocket medical fees waived, which may have contributed to catalyzing participants to seek medical care. Importantly, we included diabetes education or support trials that primarily changed the behavior of health workers within the health system, and we excluded lifestyle trials focusing on patient behavior alone without systems-level change.

Task sharing was a common thread across intervention types. Distinct from task shifting, task sharing emphasizes the shared responsibility for a task between the health workers' different levels and types of training [71]. Previous reviews of trials predominantly conducted in HICs have suggested task sharing with pharmacists as an effective strategy [8]. We found that pharmacist task-sharing interventions appeared to improve glycemic control in the pooled analysis, but the certainty of evidence was low for these types of interventions, primarily due to concerns about studies' risk of bias.

Task sharing also was a fundamental component in the nurse-led intervention by Fairall and colleagues in South Africa [29], a nurse care coordination trial [52], and multicomponent clinic-based studies [31,58,59]. We observed differences across studies relating to task sharing such as type of health worker, training, and assigned tasks. Prior reviews of task shifting for chronic diseases in LMICs have identified few trials in type 2 diabetes [72,73]. A 2019 meta-analysis by Anand and colleagues concluded that task-sharing interventions were effective in improving blood pressure in LMICs [74]. Our review shows an increase in research incorporating task sharing into health system interventions for type 2 diabetes in these settings.

Our review should be considered in the context of the movement to strengthen health systems in LMICs [75]. Diabetes has been referred to as a "tracer condition" for assessing the strength of health systems [76], and inadequate diabetes care has been reported in nationally representative surveys in many LMICs [4]. RCTs are not the only form of evidence generation in the field of health policy and research [77], and diverse research strategies are needed in conditions like type 2 diabetes that have a strong clinical evidence base yet weak evidence on implementation [3,78]. Logistical challenges in conducting randomized studies within health systems likely explain why we identified few interventions testing financial or governance

arrangements. An advantage of including only RCTs is that we are able to offer robust evidence of the impact of health system interventions on glycemic control and reveal the limited data on other outcomes. Further studies in LMICs are needed to assess non-glycemic outcomes and, given the wide prediction intervals, to determine the specific components and details of health system interventions most likely to promote effectiveness and limit potential harms. Excellent examples of implementation research include the portfolio of ongoing projects funded by the Global Alliance for Chronic Disease [79].

Our review has limitations. Defining a health system intervention is challenging, and there is no consensus definition. We justify our use of the EPOC definition as reasonable given its use in prior Cochrane EPOC reviews on health systems in LMICs. We excluded non-randomized study designs given the challenge in attributing causality for outcomes and inconsistent reporting of these designs in pilot searches. Randomized designs in health system research have limitations, including the possible attenuation of effect sizes [80]. There was statistical evidence for publication bias and substantial differences in the quality of underlying studies that limited the certainty of evidence of glycemic benefit for intervention types including pharmacist task-sharing interventions. We did not assess blood pressure outcomes given our primary interest in the evidence of interventions attempting to achieve glycemic control and prior meta-analyses supporting the effectiveness of health system interventions for blood pressure control [74,81]. The only outcome in our review for which a meta-analysis was conducted, HbA1c, is only a surrogate outcome, but it is commonly used in meta-analyses of systems-level interventions for diabetes [7,9]. Our review was restricted to studies with at least 6 months of follow-up and 100 enrolled participants. Multiple trials were included within some countries, but we did not formally account for a potential lack of independence among studies conducted within the same health system context. This aspect reflects a limitation of the evidence generated rather than one of the analysis itself. We also did not systematically assess important implementation science outcomes such as reach, fidelity, or acceptability. Finally, although there were substantial similarities within intervention types, individual studies varied by setting and population, limiting our ability to make conclusions with high degrees of certainty.

This review has notable strengths. We synthesized evidence of outcomes by focusing on RCTs and performing a meta-analysis of HbA1c. Our comprehensive search strategy facilitated this choice as we identified a larger number of trials in LMICs than previous reviews. Our review was supplemented with unpublished data received from multiple study authors, and we were able to pool HbA1c statistical estimates reported differently across studies.

In conclusion, we found that health system interventions for type 2 diabetes may be effective in improving glycemic control in LMICs, but few studies were available from rural areas or low- or lower-middle-income countries. Multicomponent clinic-based interventions had the strongest evidence for glycemic benefit among intervention types. Data were generally limited for non-glycemic outcomes such as mortality, quality of life, and cost-effectiveness. Our findings imply a need for implementation research to investigate the details of health system interventions that confer durable improvements in clinical and patient-centered outcomes in LMICs, especially in rural areas and in low- and lower-middle-income countries.

## Supporting information

**S1 Appendix. PROSPERO registration and final review protocol.**
(PDF)

**S2 Appendix. PRISMA checklist.**
(PDF)

**S3 Appendix. Search strategy.**
(PDF)

**S4 Appendix. Main domains of the EPOC taxonomy of health systems.**
(PDF)

**S5 Appendix. Characteristics of included studies.**
(PDF)

**S6 Appendix. Forest plot of overall deaths by study.**
(PDF)

**S7 Appendix. EPOC risk of bias assessment.**
(PDF)

**S8 Appendix. Overall funnel plot (all studies).**
(PDF)

**S9 Appendix. Funnel plot by intervention type (all studies).**
(PDF)

**S10 Appendix. Funnel plot generated using trim-and-fill method.**
(PDF)

**S11 Appendix. Forest plot for meta-analysis of HbA1c (%) mean difference excluding studies at high risk of bias.**
(PDF)

**S12 Appendix. Overall funnel plot excluding studies at high risk of bias.**
(PDF)

**S13 Appendix. Funnel plot by intervention type excluding studies at high risk of bias.**
(PDF)

**S14 Appendix. Forest plot generated using leave-one-out method.**
(PDF)

**S15 Appendix. Summary of findings table.**
(PDF)

**S16 Appendix. Certainty assessment of evidence for the glycemic control outcome.**
(PDF)

## Acknowledgments

We thank the following authors of included studies for contributing supplementary information used in this review: María Cecilia Anzaldo-Campos, MD, MBA; Anna Chapman, PhD; Jeroen De Man; Shaun Wen Huey Lee, PhD; Aditya Khetan, MD; Professor Dr. Anis Safura Ramli; Professor Hong-Mei Wang, PhD; and Xuefeng Zhong, MD, MPH, PhD.

The content is solely the responsibility of the authors and does not necessarily represent the official views of the funders.

## Author Contributions

**Conceptualization:** David Flood, Bradley H. Wagenaar, Elizabeth A. Rogers, Peter Rohloff.

**Data curation:** David Flood, Jessica Hane, Matthew Dunn, Sarah Jane Brown, Vineet Chopra.

**Formal analysis:** David Flood, Jessica Hane, Matthew Dunn, Bradley H. Wagenaar, Elizabeth A. Rogers, Michele Heisler, Peter Rohloff, Vineet Chopra.

**Investigation:** David Flood, Michele Heisler, Peter Rohloff, Vineet Chopra.

**Methodology:** David Flood, Jessica Hane, Sarah Jane Brown, Bradley H. Wagenaar, Elizabeth A. Rogers, Michele Heisler, Peter Rohloff, Vineet Chopra.

**Project administration:** David Flood, Jessica Hane, Matthew Dunn, Vineet Chopra.

**Resources:** David Flood, Sarah Jane Brown.

**Software:** David Flood, Sarah Jane Brown.

**Supervision:** David Flood, Peter Rohloff, Vineet Chopra.

**Visualization:** David Flood, Matthew Dunn.

**Writing – original draft:** David Flood, Peter Rohloff, Vineet Chopra.

**Writing – review & editing:** David Flood, Jessica Hane, Matthew Dunn, Sarah Jane Brown, Bradley H. Wagenaar, Elizabeth A. Rogers, Michele Heisler, Peter Rohloff, Vineet Chopra.

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
