## [Editor Report · Decision Letter 0]

4 May 2020

Dear Dr Flood, 

Thank you for submitting your manuscript entitled "Health system interventions for adults with type 2 diabetes in low- and middle-income countries: A systematic review and meta-analysis" for consideration by PLOS Medicine.

Your manuscript has now been evaluated by the PLOS Medicine editorial staff and I am writing to let you know that we would like to send your submission out for external peer review.

Kind regards,

Artur Arikainen,

Associate Editor

PLOS Medicine

---

## [Decision Letter · Decision Letter 1]

7 Jul 2020

Dear Dr. Flood,

Thank you very much for submitting your manuscript "Health system interventions for adults with type 2 diabetes in low- and middle-income countries: A systematic review and meta-analysis" (PMEDICINE-D-20-01702R1) for consideration at PLOS Medicine. 

Your paper was evaluated by a senior editor and discussed among all the editors here. It was evaluated by four independent reviewers, including a statistical reviewer. The reviews are appended at the bottom of this email and any accompanying reviewer attachments can be seen via the link below:

[LINK]

In light of these reviews, I am afraid that we will not be able to accept the manuscript for publication in the journal in its current form, but we would like to consider a revised version that addresses the reviewers' and editors' comments. Obviously we cannot make any decision about publication until we have seen the revised manuscript and your response, and we plan to seek re-review by one or more of the reviewers. 

We expect to receive your revised manuscript by Jul 28 2020 11:59PM. Please email us (plosmedicine@plos.org) if you have any questions or concerns.

We look forward to receiving your revised manuscript. 

Sincerely,

Emma Veitch, PhD

PLOS Medicine

On behalf of Clare Stone, PhD, Acting Chief Editor,

PLOS Medicine

plosmedicine.org

*In the last sentence of the Abstract Methods and Findings section, please add a brief note summarising any key limitation(s) of the study's methodology.

*At this stage, we ask that you include a short, non-technical Author Summary of your research to make findings accessible to a wide audience that includes both scientists and non-scientists. The Author Summary should immediately follow the Abstract in your revised manuscript. This text is subject to editorial change and should be distinct from the scientific abstract. Please see our author guidelines for more information: https://journals.plos.org/plosmedicine/s/revising-your-manuscript#loc-author-summary

Comments from the reviewers:

Reviewer #1: This is an important but challenging area to investigate. The authors have done an excellent job of bringing together a large number of trials. The challenge is in the interpretation of findings when the results of studies are grouped and meta-analysed. The large I-squared values indicate the statistical risks in pooling these studies. This is compounded by the obvious heterogeneity of study design. Even within the broad categories of study type, there are unlikely to be two studies whose description of their intervention was the same, never mind how and how successfully the protocol was implemented. Added to this is the variation in the setting and populations, which would likely impact on the potential for and size of any benefit. Thus, it is quite likely that within the studies in each group there are interventions that work and some that don't. Indeed, it is likely that the need for some studies was justified on the basis that they were attempting to improve on other similar interventions that failed. In such circumstances it is very hard to know what a pooled estimate means, or even how to interpret general comments about the benefits of, for example, multicomponent interventions. It seems that it would be more useful to try to understand what features characterized the studies with the larger effect sizes. Of course this is also inherently difficult, as conclusions may then be driven by very small numbers of studies.

Ultimately, it seems that with current methodologies, it is very hard to make robust conclusions in this area.

Reviewer #2: See attachment

Michael Dewey

Reviewer #3: This systematic review and meta-analysis achieves its intended aims and has been conducted with sufficient rigour. 

The introduction provides a clear rationale of the need for this study. It may be of benefit to include example/s of some health service interventions within the introduction (paragraph 2). While the EPOC definition is stated within the methods section, some examples earlier in the introduction would assist readers to interpret the distinction between clinical interventions and health service interventions. 

The methods section provides sufficient detail as per the PRISMA recommendations. Information is provided with regard to eligible interventions, however, eligible control conditions do not appear to be stated throughout the manuscript. Given the intention of a meta-analysis is to quantify the effects of a specific intervention in relation to a specified control condition, this is an important consideration to include throughout the entire manuscript. This would also assist the reader to determine if the meta-analysis is comparing similar studies. It is possible that the heterogeneity observed may have been due to studies using different control conditions. 

The results section provides a excellent synthesis of both the narrative and quantitative components. When summarizing outcomes, I would find it of interest to state how many of the included studies had a primary outcome of HbA1c (line 195). 

The discussion section solidly interprets the findings in relation to previous literature and acknowledges the strengths and limitations of the review. 

Overall a well written, well conducted, interesting piece of work.

Reviewer #4: Dr Flood and coworkers have conducted a systematic review and meta-analysis of reports (trials) on health system intervention for diabetes control in LIMC, using change in HbA1c as main outcome. After extensive searchers they identified 38 eligible studies, of which 34 were included in meta-analysis. They found a significant effect of the intervention with a change in HbA1c of 0.46% overall, 0.37% in multi-component clinic-based interventions, 0.93% in pharmacist task-sharing studies, and -0.27% in trials of diabetes education and support alone. Other pre-specified outcomes were inconsistently reported. They concluded on a likely efficacy of health interventions and advocated for further studies to assess the impact on non-glycaemic outcomes.

The review addresses a question of global relevance, the manuscript is generally well written and easy to read. I have the following suggestions to the authors.

1) There seems to be suggestions that non-English language studies were excluded in the selection process with 20 such studies excluded in the second last stage. Should some of these studies be eligible, then it will be a bias and the authors should rather consider using translated to access the information in non-English language studies for possible inclusion in the review.

2) Did the investigators make effort to contact the primary investigators for missing data; there seems to be indications that soem studies were excluded for missing data.

1) The authors have identified possible publication bias from funnel plots and have tried to explore this further using subgroup analyses. However, because of the overall small number of studies included, subgroups analyses are likely to be under powered to reliably explore publication bias. I am advising the authors to consider using the 'trim and fill' methods to impute the missing studies, and see if this results in imputed studies with plausible effect size. Similarly the authors should consider running other sensitivity analyses to confirm the robustness of their findings. I am thinking here of the 'leave one out' approach for instance, to check if some particular studies had more effect on the overall estimates.

[LINK]

---

## [Decision Letter · Decision Letter 2]

25 Aug 2020

Dear Dr. Flood,

Thank you very much for submitting your manuscript "Health system interventions for adults with type 2 diabetes in low- and middle-income countries: A systematic review and meta-analysis" (PMEDICINE-D-20-01702R2) for consideration at PLOS Medicine. 

Your revised paper was evaluated by a senior editor and discussed among all the editors here. It was also sent to three of the four original independent reviewers once more, including a statistical reviewer. The reviews are appended at the bottom of this email and any accompanying reviewer attachments can be seen via the link below:

[LINK]

In light of these reviews, I am afraid that we will still not be able to accept the manuscript for publication in the journal in its current form, but we would like to consider another revised version that addresses the reviewers' and editors' comments. Obviously we cannot make any decision about publication until we have seen the revised manuscript and your response, and we plan to seek re-review by one or more of the reviewers. 

We expect to receive your revised manuscript by Sep 15 2020 11:59PM. Please email us (plosmedicine@plos.org) if you have any questions or concerns.

We look forward to receiving your revised manuscript. 

Sincerely,

Artur Arikainen

Associate Editor

PLOS Medicine

plosmedicine.org

1. Please address all of the reviewers’ comments below. Notably, we request that you update your study to include non-English studies and the additional sensitivity analyses requested. We appreciate that the inclusion of non-English language studies may require additional expense and effort, but we agree with reviewer #4 that the review would be incomplete without these, particularly given the focus on LMICs.

2. Abstract:

a. Please report all of the elements required by PRISMA for abstracts: http://www.plosmedicine.org/article/info:doi/10.1371/journal.pmed.1001419 . Specifically, please include a description of how study quality/risk of bias was assessed, and what the overall quality/risk of bias was found to be.

b. Please give the total number of patients across all included studies.

c. Please clarify (here and in the main Methods) that the search dates were “from inception through February 24, 2020”, as appropriate.

d. Please include a breakdown of the number of each study type for the included studies (eg. “18 cluster RCTs, 20 individual RCTs”), and regions covered (as from table 1).

e. Please include line numbers in the margin throughout.

3. Page 25: Please remove the Competing Interests, Data Availability, Funding and Author Contribution statements – these should be filled in on the online submission form instead. Please keep the Acknowledgements here as they are.

4. When completing the PRISMA checklist, please use section and paragraph numbers, rather than page numbers. Your checklist also appears to be cut off after item 14 – please include the full checklist.

Comments from the reviewers:

Reviewer #1: I think that the reviewers have done an adequate job in responding to my initial concerns.

The authors conclude that the most robust evidence of benefit is for multi-component interventions. However, this seems at odds with the data in Fig 2. Pharmacist task sharing interventions include more trials (albeit with fewer participants), a larger point estimate for the effect size, and 8/12 trials with individually statistically significant benefits. It appears that the downgrading of the Pharmacist trials is based on the GRADE score in Appendix S12, which is rated as 2/4 for the pharmacy trials compared to 3/4 for the multi-component trials. No discussion is provided on what the actual difference is between the quality scores, though this is central to the main study conclusion. The relevance of this is not only in regard to whether or not the evidence supports pharmacist task sharing interventions, but whether or not single-component interventions in general might be effective. Several other studies in this setting have concluded that multi-component interventions are required to be effective, and the current presentation of this meta-analysis supports this. However, that conclusion turns out to be entirely dependent on a difference in GRADE scores that is not explained or discussed. 

Reviewer #2: The authors have addressed my points but I still have a couple of things to raise.

Even is the authors do not wish to do a formal analysis of mortality and the other outcomes I think expecting the reader to work all through S5 is unnecessary when they could be presented by outcome. I would put them in a forest plot but if the authors wish they can suppress the summary estimates.

I see there is some divergence of opinion between the referees about the heterogeneity and possible small study effects. For the record I have no problem with doing a summary in the presence of extreme statistical heterogeneity as measured by I^2. Whether the clinical heterogeneity precludes it is not for me to say. One thing I should have suggested and I apologise for not having done this at the beginning is that prediction intervals in addition to confidence intervals would be useful. They give a picture of where the next study would be likely to fall which I would imagine is more useful to someone working in public health in a different country than knowing the properties of the distribution of the summary statistic. Riley et al have argued for the use of prediction intervals https://doi.org/10.1136/bmj.d549 and it is not super-difficult to calculate them even if the authors' software does not do it. In the presence of the heterogeneity I do not think it is worth spending much time on possible small study effects and trim and fill has rather fallen out of favour recently.

I think I failed to explain adequately my concern about lack of independence of studies within a country. It is not just a concern about the same research team being involved which I am sure the authors have checked but also the fact that studies in a single country must share the same background health system and so will be more similar than those from a different country. I do not think country needs inclusion in a formal analysis but I would mention this as a possible limitation.

Michael Dewey

Reviewer #4: Thanks for the opportunity of reading the revised manuscript.

I note however that the authors have basically declined answering my two major comments:

1) They seem to be unable to access appropriate translation service to enable them assessing all potentially eligible studies; and therefore have limited inclusion to only studies published in English. In doing so, they have excluded at least 20 studies from further assessment due to language issue. Not only this deliberately introduces publication bias, but I am not sure if it would be acceptable for robust systematic review. The authors should consider the option of accounting at least for few other major international languages.

2) I proposed that they use sensitivity analyses (trim and fill) to explore the effect of missing studies imputation on the publication bias. I am not convinced by the rebuttals of the authors. Should the sensitivity analyses show that imputed studies have plausible effect sizes, then the reader needs to know, since accounting for missing studies could completely change the overall estimates from the analyses conducted by the authors. I also advised the authors to run other sensitivity analyses (leave one out) to confirm the robustness of their findings, what they haven't done.

[LINK]

---

## [Decision Letter · Decision Letter 3]

2 Oct 2020

Dear Dr. Flood,

Thank you very much for re-submitting your manuscript "Health system interventions for adults with type 2 diabetes in low- and middle-income countries: A systematic review and meta-analysis" (PMEDICINE-D-20-01702R3) for review by PLOS Medicine.

I have discussed the paper with my colleagues and the academic editor and it was also seen again by three reviewers. I am pleased to say that provided the remaining editorial and production issues are dealt with we are planning to accept the paper for publication in the journal.

[LINK]

We look forward to receiving the revised manuscript by Oct 09 2020 11:59PM. 

Sincerely,

Artur Arikainen, 

Associate Editor 

PLOS Medicine

plosmedicine.org

Requests from Editors:

1. Short title: Please amend to: “Systematic review of health system interventions for adults with type 2 diabetes in low- and middle-income countries”

2. Abstract: 

a. Please include a summary breakdown of included studies by region.

b. Lines 65-68: Please include p values for each result.

c. Around line 69: Please include this sentence from lines 362-363: “Eight studies were at low risk of bias for the summary assessment of glycemic control, 15 studies were at unclear risk, and 16 studies were at high risk.”

d. Line 76: Please begin with: “In this meta-analysis, we found that…”

3. Author summary: Line 94: Please define “glycemic control” for a lay reader, particularly as you mention “blood glucose control” later.

4. Line 213: Please remove this section on Funding. Please include this sentence elsewhere in the methods: “Ethical approval was not required as the research used publicly available data.”

5. Please remove spaces from within citation callouts throughout, eg “…professionals [30,31], and in…” (line 241).

6. You state that the multicomponent clinic-based interventions had the "most supporting evidence" (lines 78, 99-100, and 509-510). Yet there were apparently more pharmacist task sharing intervention trials included, and in these studies there was a stronger effect on the glycemic endpoint. It looks like the clinic-based studies were larger and higher quality, though. We would ask you to consider adapting the phrase "most supporting" to explicitly refer to strength of evidence, eg “…had the strongest evidence for…”.

Comments from Reviewers:

Reviewer #1: I am satisfied with the responses.

Reviewer #2: The authors have addressed all my points.

Michael Dewey

Reviewer #4: The authors have addressed all my comments.

[LINK]

---

## [Editor Report · Decision Letter 4]

19 Oct 2020

Dear Dr. Flood, 

On behalf of my colleagues and the academic editor, Dr. Andre P Kengne, I am delighted to inform you that your manuscript entitled "Health system interventions for adults with type 2 diabetes in low- and middle-income countries: A systematic review and meta-analysis" (PMEDICINE-D-20-01702R4) has been accepted for publication in PLOS Medicine. 

PRODUCTION PROCESS

Before publication you will see the copyedited word document (within 5 business days) and a PDF proof shortly after that. The copyeditor will be in touch shortly before sending you the copyedited Word document. We will make some revisions at copyediting stage to conform to our general style, and for clarification. When you receive this version you should check and revise it very carefully, including figures, tables, references, and supporting information, because corrections at the next stage (proofs) will be strictly limited to (1) errors in author names or affiliations, (2) errors of scientific fact that would cause misunderstandings to readers, and (3) printer's (introduced) errors. Please return the copyedited file within 2 business days in order to ensure timely delivery of the PDF proof. 

If you are likely to be away when either this document or the proof is sent, please ensure we have contact information of a second person, as we will need you to respond quickly at each point. Given the disruptions resulting from the ongoing COVID-19 pandemic, there may be delays in the production process. We apologise in advance for any inconvenience caused and will do our best to minimize impact as far as possible.

PRESS

PROFILE INFORMATION

Thank you again for submitting the manuscript to PLOS Medicine. We look forward to publishing it. 

Best wishes, 

Artur Arikainen, 

Associate Editor 

PLOS Medicine

plosmedicine.org